# Temozolomide Efficacy and Metabolism: The Implicit Relevance of Nanoscale Delivery Systems [note 1]

**DOI:** 10.3390/molecules27113507

**Published:** 2022-05-30

**Authors:** Daria Petrenko, Vladimir Chubarev, Nikita Syzrantsev, Nafeeza Ismail, Vadim Merkulov, Susanna Sologova, Ekaterina Grigorevskikh, Elena Smolyarchuk, Renad Alyautdin

**Affiliations:** 1Department of Pharmacology, Sechenov University, 119019 Moscow, Russia; tchoubarov@mail.ru (V.C.); nicksyz13@gmail.com (N.S.); merkulov_v_a@staff.sechenov.ru (V.M.); susanna.sologova@yandex.ru (S.S.); grigorevskikh_e_m@staff.sechenov.ru (E.G.); smolyarchuk_e_a@staff.sechenov.ru (E.S.); 2Department of Pharmacology, University Technology MARA, Kuala Lumpur 50450, Malaysia; nafeeza06@hotmail.com; 3Scientific Centre for Expert Evaluation of Medicinal Products, 127051 Moscow, Russia

**Keywords:** glioblastoma, temozolomide (TMZ), MTIC, nanocarriers, drug delivery systems, nanoparticles (NP)

## Abstract

The most common primary malignant brain tumors in adults are gliomas. Glioblastoma is the most prevalent and aggressive tumor subtype of glioma. Current standards for the treatment of glioblastoma include a combination of surgical, radiation, and drug therapy methods. The drug therapy currently includes temozolomide (TMZ), an alkylating agent, and bevacizumab, a recombinant monoclonal IgG1 antibody that selectively binds to and inhibits the biological activity of vascular endothelial growth factor. Supplementation of glioblastoma radiation therapy with TMZ increased patient survival from 12.1 to 14.6 months. The specificity of TMZ effect on brain tumors is largely determined by special aspects of its pharmacokinetics. TMZ is an orally bioavailable prodrug, which is well absorbed from the gastrointestinal tract and is converted to its active alkylating metabolite 5-(3-methyl triazen-1-yl)imidazole-4-carbozamide (MTIC) spontaneously in physiological condition that does not require hepatic involvement. MTIC produced in the plasma is not able to cross the BBB and is formed locally in the brain. A promising way to increase the effectiveness of TMZ chemotherapy for glioblastoma is to prevent its hydrolysis in peripheral tissues and thereby increase the drug concentration in the brain that nanoscale delivery systems can provide. The review discusses possible ways to increase the efficacy of TMZ using nanocarriers.

## 1. Introduction

Primary malignant and benign tumors of the CNS (CNS PTs) account for about 2% of all human tumors, or, according to CBTRUS (The Central Brain Tumor Registry of the United States), 23.41 cases per 100,000 population. Among the CNS PTs, meningiomas (35.6%, with only 1% of them being malignant) and gliomas (35.5%, with 14.6% of the total number of primary brain tumors being glioblastomas) predominate. Pituitary tumors account for 15%, neuromas of the eighth nerve—for 8% of the total number of brain tumors [1,2,3].

The most common primary malignant brain tumors in adults are gliomas, which account for 81% of all malignant brain tumors. They can develop anywhere in the central nervous system, but are mainly found in the brain and glial tissue [1,4]. Although these tumors are usually malignant, some types do not always behave as malignant ones. It is the second most common form of cancer (after leukemia) in children and teenagers [5].

Gliomas originate from astrocytic, oligodendrocytic, or a mixture of these two types of cells, and are usually classified according to the International Classification of Cancer Version 3 (ICD-O-3) and the World Health Organization (WHO) Grade [6]. Grades I–IV of the WHO classification are characterized by malignant behavior. The most common histological types of gliomas in adults include astrocytoma (grades I–IV), oligodendroglioma (grades II–III), and oligoastrocytoma (grades II–III). However, there is no consensus on gliomas as a larger class of histology, which may complicate comparisons between studies [1,6]. This review primarily covers the recent (2002–2013) studies of selected risk factors in the epidemiology of glioma in adults (age ≥ 20 years), with the exception of ependymomas, which are extremely rare [7].

Patient survival after diagnosis of glioma varies widely. Thus, oligodendroglioma, a diffuse infiltrative and highly differentiated glioma built from tumor cells morphologically similar to oligodendroglia, has increased the chances of patient survival when accompanied by the 1p/19q deletion and absence of P53 expression [8]. Prognostic factors for patients with glioma are: age (after 40 years, the prognosis decreases with age), score on the Karnofsky scale, tumor volume, tumor localization in vital structures, and foci of contrast accumulation in the tumor according to the MRI of the brain [9,10]. According to Wanis et al. (2021), between 1995 and 2017, 133,669 cases of primary brain tumors in adults were recorded in England. Glioblastoma was the most common tumor subtype (31.8%), followed by meningioma (27.3%). The age-standardized incidence of glioblastoma increased from 3.27 per 100,000 population per year in 1995 to 7.34 in men in 2013 and from 2.00 to 4.45 in women. The incidence of meningioma also increased from 1.89 to 3.41 per 100,000 in men and from 3.40 to 7.46 in women. The incidence of other astrocytic and unclassified brain tumors decreased between 1995 and 2007 and has remained stable thereafter. This increase in the incidence of brain tumors can be attributed to several reasons. The most probable factor is the improvement in the diagnosis of brain tumor diseases due to the use of new diagnostic tools, molecular tests, as well as the optimization of organizational and statistical methods of oncological service registration [11]. The increase in the incidence of glioblastoma, the most aggressive form of malignant brain tumor, makes the task of adequate effective treatment extremely urgent. Despite the emergence of new effective anticancer drugs, the main problem remaining is overcoming the blood–brain barrier (BBB). Such a complex system of clear and strict sorting of substances that pass the central nervous system creates difficulties in the application of therapy aimed at the targets of the central nervous system. To obtain a therapeutic effect, it is necessary to reach a specific target zone, sufficient concentration of the drug, the duration of its action, and safety for the surrounding organs and tissues. For many years, scientists have been developing a strategy for vector delivery of therapeutic molecules by overcoming the selective barrier and simultaneously preserving the drug potential and safety profile. Nano-transporters are being actively studied as such carriers [12]. The study and use of nanotechnologies for the creation of medicines is currently a promising and rapidly developing area of medical chemistry and pharmacology. The use of nanoscale medicinal products makes it possible to optimize early diagnosis of diseases, ensure targeted delivery of drugs to target organs, and increase the effectiveness of therapy compared to existing treatment methods. The world’s first nanopreparation—Doxil^®^ (liposomal pegylated form of doxorubicin with reduced toxicity)—was registered in 1995 in the USA. By 2019, the volume of investments in the development of nanopreparations is estimated to be USD178 billion. More than a hundred nanopreparations with various indications have been introduced to the pharmaceutical market: chemotherapeutic drugs doxorubicin, paclitaxel (Abraxane^®^), antifungal drugs amphotericin B, etc. Sales of the drug Abraxane^®^ for several oncological indications in 2015 were estimated at USD967 million [13,14,15,16].

The dimensional characteristics of nanopreparations, entailing changes in physicochemical parameters (including solubility, reactivity), as well as pharmacokinetic parameters (bioavailability, absorption, distribution, metabolism, and excretion) require careful the study of the effect on the benefit/risk ratio when using them. Factors that increase the risk of using certain nanopreparations, for example, insufficient efficacy or severity of side effects, may lead to the termination of clinical trials or even the circulation of such drugs [17,18].

The following types of nanoscale drugs and their carriers are distinguished, the most commonly used in clinical practice:(1)Inorganic nanoparticles (Fe_3_O_4_, TiO_2_, ZnO, SiO_2_, etc.);(2)Liposomes;(3)Polymer–drug, polymer–protein conjugates;(4)Micelles (block copolymer micelles);(5)Nanocrystals;(6)Viral carriers of drugs.

Inorganic nanoparticles, in particular, metal oxide nanoparticles, are among the most common types of nanomaterials currently being studied and used. Iron oxide (Fe_3_O_4_) and silicon dioxide (SiO_2_) are widely used for therapeutic purposes, including targeted drug delivery. Iron oxide (Fe_3_O_4_) and rare earth element complexes, in particular gadolinium (Gd), are used for diagnostic purposes as contrasting agents for magnetic resonance imaging (MRI). Zinc oxide (ZnO) nanoparticles are used in the production of sunscreens, food additives, pigments, and biosensors. The nanoscale form of titanium dioxide (TiO_2_) is used as a pigment, thickener, and absorber of ultraviolet rays in cosmetics and skin care products. In addition, these nanoscale particles are used to facilitate the integration of artificial medical implants into bone tissue.

Liposomes are lipid vesicles consisting of one or more phospholipid bilayers separated by an aqueous phase, used for targeted drug delivery. This structure was discovered in the mid-1960s of the last century when studying the role of phospholipids in blood clotting. Liposomes are one of the most common types of nanocarriers used in systems for targeted drug delivery. In addition, so-called “fliposomes” liposomes have been developed with substances embedded in the lipid bilayer—molecular switches that react, for example, to changes in the pH of the medium or temperature conditions in a given area. As a result of the changes in external conditions (pH or temperature), a conformational transition of the molecular switch occurs, which leads to the rupture of the lipid bilayer and the release of the encapsulated substance. Currently, clinical trials of thermosensitive liposomes with doxorubicin (ThermoDox^®^) are being conducted, releasing the active substance during local heating of tissues by ultra-high frequency radiation [19,20,21].

Polymer–drug or protein–drug conjugates are molecules of the active substance of a drug associated with polymer or protein structures. Conjugation with macromolecular carriers changes the rate of drug excretion from the body and provides the possibility of its release over a long period of time. In addition, it limits the absorption of the drug by cells during endocytosis, thereby contributing to targeted delivery. Since highly soluble proteins and polymers are mainly used as a carrier, the use of such conjugates also makes it possible to solve the problem of solubility of low-soluble drugs.

Micellar nanoscale drugs are colloidal block copolymer amphiphilic nanoscale carriers with encapsulated active substance. Polymer micelles often serve as carriers of hydrophobic compounds, improving the solubility of the drug. The use of nanocarriers with a relatively high molecular weight ensures a longer circulation of the drug in the blood. When the drug is delivered to the target tissue, the micelles are able to disintegrate into lower molecular structures, which makes it possible to remove the surfactants that make up the micelle from the body [22,23,24].

Nanocrystals are crystals whose dimensions in one or more dimensions lie in the nanoscale range. Nanocrystalline drugs allow us to obtain the advantages associated with the transition to the nanoscale range (improved solubility, bioavailability), compared with macroanalogs without the use of excipients.

The most common nanoscale carriers based on virions are retroviruses, adenoviruses, and adeno-associated viruses. The main advantage of viral vectors is their high efficiency as a tool for delivering genetic material to tissue cells, as well as greater biocompatibility and biodegradability compared to synthetic nanomaterials. Thus, in the preparations, alipogene typarvovec (Glybera) of UniQure and voretigene neparvovec(Luxturna) of Novartis intended for the treatment of diseases caused by an inherited defect (mutation) of a gene in the patient’s body, the delivery of a working copy of the genes is carried out using an adeno-associated viral vector [22,25,26,27].

## 2. Treatment Options of Glioblastoma

Currently, the standards for the treatment of glioblastoma include a combination of surgical, radiation, and drug therapy methods [28,29].

In surgical treatment, diffuse infiltration of invasive GBM cells, which prevents the complete resection of cancerous tissue, makes tumor recurrence almost inevitable. That is why pharmacotherapy is currently a necessary component of GBM combination therapy. The drug therapy currently includes temozolomide (TMZ), an alkylating agent, and bevacizumab, a recombinant monoclonal IgG1 antibody that selectively binds to and inhibits the biological activity of vascular endothelial growth factor. According to Stupp et al. (2005), the supplementation of glioblastoma radiation therapy with TMZ increased patient median survival from 12.1 to 14.6 months [30,31]. These results indicate the possibility of slowing down glioblastoma development by means of drug therapy. However, the severity of the condition and the efficacy characteristics of existing drugs indicate the need to search for fundamentally new methods of drug therapy for glioblastoma [32,33,34].

The main obstacle to expanding the range of anticancer drugs for the successful pharmacological treatment of tumors of the central nervous system (CNS) is the presence of the BBB, which prevents the passage of most anticancer agents, both hydrophilic and lipophilic, into the CNS [35,36,37]. Hydrophilic compounds leave the vascular bed of the capillaries of peripheral tissues due to paracellular transport, which is not possible in the capillaries of the cerebral vessels because of existence of tight junctions between endothelial cells. Many lipophilic compounds that penetrate into the BBB-forming endotheliocytes through the luminal bilayer become a substrate for P-glycoprotein (Pgp; MDR1; ABCB1) expressed in the apical membrane of these cells. TMZ is one of the few anticancer drugs capable of overcoming multiple BBB obstacles [38,39,40].

## 3. Temozolomide (TMZ): Efficacy and Metabolism

TMZ is a potent radiosensitizer and a key component of chemoradiation therapy for patients with newly diagnosed glioblastoma. The main advantage of temozolomide is its high effectiveness against malignant glioma of the brain, the main disadvantage is the safety profile. TMZ is a very toxic drug for surrounding tissues; therefore, nanoparticles are considered as promising carriers capable of leveling this disadvantage, while increasing penetration through the blood–brain barrier and reducing the metabolic rate of TMZ. The efficacy of TMZ for the treatment of GBM is limited. This is partly due to the high level of DNA repair activity of O-6-methylguanine-DNA methyltransferase (MGMT) in tumor cells, which transfers the methyl group to the internal cysteine acceptor residue and reduces the effect of the alkylating agent. Unfortunately, other established mechanisms contribute significantly to the development of TMZ resistance (overexpression of epidermal growth factor receptor (EGFR), galectin-1, murine double minute 2 (Mdm2), and p53 gene mutation [41,42,43,44]. A promising way to increase the TMZ efficacy is to include it in nanostructures that can protect TMZ in the blood and penetrate through the BBB. As stated above, TMZ is stable in acidic medium, but at neutral pH or pH > 7 it decomposes to active metabolite MTIC [45,46]. In contrast, MTIC is stable in alkaline medium, but rapidly fragments in a methylating mode at pH < 7 [43,46]. Thus, TMZ, but not MTIC, is able to cross the BBB to the brain, where it will be hydrolyzed to MTIC [47,48]. An increase in the half-life of TMZ in the blood under physiological conditions will result in greater accumulation of TMZ in the brain, and, as a consequence, in the GBM region, before its transformation to MTIC. Consequently, the drug will be more efficacious, and lower doses could be used to maintain the current TMZ therapeutic level. This would make chemotherapy more optimistic and reduce side effects and toxicity, increasing patients’ lifespans [45,46].

In order to protect the prodrug (TMZ) from degradation in peripheral tissues, polymer complexes can be used to limit its contact with surrounding tissues.

## 4. Nanoscale Delivery Systems

The idea of drug delivery with the help of nanocarriers originated at the beginning of the 20th century, and in 1974, Professor Gregory Gregoriadis proposed the inclusion of therapeutic substances into liposomes, which paved the way for the use of liposomal nanocontainers. Over the past few decades, there has been a breakthrough in the field of targeted transportation of pharmaceutical substances via nanotechnology. At present, there are many multifunctional forms of nanoparticles that can deliver a medicine, such as temozolomide, to certain target cells. TMZ enclosed in a nanocarrier can overcome the BBB, while exerting less toxicity on healthy tissues [49,50,51]. Nanotransporters can be divided into classes that differ in their physical, chemical, and biological properties:∗Metal nanoparticles (gold, silver, zinc oxide);∗Polymer carriers (polymer micelles, polymer nanoparticles–chitosan, polymethylacrylate; nanospheres, nanocontainers, nanoglobules, dendrimers, nanoconjugates, polymerosomes);∗Carbon nanotubes;∗Quantum dots;∗Lipid systems (solid lipid nanocarriers, cationic/anionic liposomes);∗Nanoemulsions (cubosomes of 50–700 nm), etc.

In this review, we consider the advantages and disadvantages of various nanoparticle systems, the possibility of including TMZ in their structure, experimental evidence of using this antitumor agent as part of nanotransporters, and the effectiveness of this approach as compared to the free form of delivery (Figure 1) [52].

### 4.1. Liposomes

Among all other lipid nanocarriers, liposomes have the greatest potential for clinical use. They are small spherical vesicles consisting of cholesterol and a phospholipid bilayer that closes on itself. Thus, a hollow shell structure with an aqueous core is formed, which can encapsulate both hydrophilic and lipophilic drugs. The lipid layer prevents the degradation of the encapsulated drug, and also protects the surrounding tissues from its effects [51,52,53]. Liposomes have exceptional mechanical strength and flexibility, similar to lysosomes, which account for their low toxicity, good biocompatibility, and a tendency to physiological decomposition. The most suitable method suggested by scientists to obtain liposomes loaded with temozolomide (TMZ-liposomes) is reverse phase evaporation (REV), since TMZ is thermosensitive (decomposes above 45 °C) [52]. This method allows for capturing of a high dose of the substance (54.02 ± 2.001%) and ensures its long-term release. It is noteworthy that the ability of TMZ to cause hemolysis of red blood cells is eliminated in the composition of liposomes [52].

To protect liposomes from phagocytosis in the reticuloendothelial system, their surface is modified with hydrophilic polymers, such as polyethylene glycol—the process called pegylation. Pegylated liposomes accumulate in the blood and brain, and the drug concentration increases by 1.6 and 4.2 times, respectively [53] (Figure 2).

M. Gabay et al. reported that the conjugation of a TMZ liposome with a brain-specific peptide (1,2-dioleoyl-sn-glycero-3-succinate-Cys-His-Leu-Asp-Ile-Ile-Trp-COOH) provides a greater passage of the transport system through the BBB than a free drug. Liposomal targeted therapy with temozolomide demonstrated a significant increase in survival in mice, demonstrating a life expectancy increase by 62%. This approach makes it possible to reduce the dose of the drug and suppress the development of drug resistance [54,55,56,57,58,59].

Despite the increased effectiveness in comparison with traditional methods of drug delivery, and the promising properties of liposomes, there is still no FDA-approved clinical liposomal product for the treatment of CNS pathologies. This is due to spontaneous factors that cannot be embraced by studies [51,56,57].

It is worth mentioning the method of direct infusion of a medicinal substance under controlled pressure by placing a microscopic catheter in the area of the brain tumor—convection-enhanced delivery (CED). TMZ encapsulated in liposomes is purposefully injected into the interstitial fluid of the brain, leading to increased TMZ accumulation in the tumor focus and decreased systemic toxicity of the drug. This strategy gives a 100-fold increase in the intraparenchymal concentration of drugs, as compared to the intranasal and intravenous methods of administration [59]. The physicochemical characteristics of liposomes affect the degree of absorption of drugs and the nature of their release. The pharmacological activity of these nanotransporters also depends on their surface charge: the cationic surfaces of liposomes easily interact with anionic charged cell membranes due to greater affinity [53,60,61]. This leads to the accumulation of the drug and positive protons in the lysosomes of cells, their acidification and further lysis with the release of lysosomal content [62,63,64]. Thus, pegylated liposomes are the most effective ones, since they are better protected from endogenous degradation thanks to polyethylene glycol. The most promising administration method is invasive direct infusion, enhanced by convection. It should be added that the ability of liposomes to aggregate and hydrolyze can be eliminated by creating proliposomes that differ in their smaller size, greater stability, and solubility [58,59,65,66]. However, this class of nanotransporters has not been sufficiently studied yet, and further research is required in this field for their full-fledged internationally approved clinical use.

### 4.2. Solid Lipid Nanoparticles and Nanostructured Lipid Carries

Other types of nanotransporters are solid lipid nanoparticles (SLNs) and nanostructured lipid carriers (NLCs) [66,67]. SLNs are aqueous colloidal dispersed particles consisting of a solid lipid core stabilized by surfactants. They are non-toxic, biodegradable, but are inferior to NLCs in terms of drug encapsulation efficiency and stability. NLCs are a new generation of SLNs, in which the solid core is replaced by the liquid one [68,69]. Given the high lipophilicity of TMZ, the researchers suggested that NLCs would be a more acceptable system for transporting the drug due to its expected greater solubility in the liquid core. The described advantages made it possible to significantly increase the amount of substance encapsulated by NLC transporters [66,70]. According to experimental data, temozolomide contained in solid lipid nanoparticles (TMZ-SLN) is less effective than that enclosed in the structure of nanolipid carriers (TMZ-NLC) [71,72]. SLNs included stearic acid and soy lecithin (stearic acid and soya lecithin), NLCs were made by modifying SLNs using polyoxyl castor oil and soy phosphatidylcholine (Cremophor ELP and SPC). NLCs outperformed SLNs in all parameters: NLC size was only 120 nm, compared to larger SLNs (180 nm); TMZ capture reached 90% in the case of NLCs and only 80% in SLNs; the IC50 value was less in NLCs—0.55 microns (compared to 5.26 microns in SLNs); in addition, an exceptionally long TMZ release of 48 h was observed in the NLC system. Based on the data obtained, the scientists concluded that TMZ-NLC was the preferred option in the treatment of glioblastoma. Furthermore, the addition of chitosan to nanolipid carriers increased the diffusion of the drug through the mucous membranes during intranasal administration, because of the interaction between the positive charge of the chitosan amino group and the negative charge due to salicylic acid of epithelial cell membranes [73,74]. This makes it possible to increase TMZ accumulation in the required region and, therefore, to reduce the dose of the loaded drug and, consequently, its toxicity [51,75].

### 4.3. Polymeric Nanoparticles

Polymeric nanoparticles (PNPs) and copolymer-based nanocarriers have proven to be promising methods of TMZ delivery. These are solid colloidal particles ranging in size from 100 to 200 nm, capable of targeted transportation of the captured drug into the tumor [76,77,78]. These systems can be obtained naturally and synthetically, and one of them—a polymer based on lactic-co-glycolic acid (PLGA)— is approved by the USFDA. Temozolomide was loaded into a PLGA-NP, then the resulting prefabricated particles were placed in a thermoreversive hydrogel system, which contributed to the manifestation of thixotropic properties of nanoparticles. The sol–gel transition is aimed at a the well-controlled release of the drug by increasing TMZ half-life. The action of PNPs has also been demonstrated in the work by Tian et al. by noninvasive enhanced delivery of TMZ to the brain [30]. The researchers used a biodegradable, minimally toxic polymer polybutyl cyanoacrylate (PBCA) due to its biocompatibility. When modified with polysorbate 80 surfactant by emulsion polymerization, more “mobile” PBCA-TMZ NPs with a size of 135 ± 11 nm were obtained. The authors compared the release of NPs without the drug and NPs loaded with TMZ. The following results were received: the effective encapsulation (EE) was 44.23 ± 2.04% and drug loading was 2.80 ± 0.05%. Despite the release of 80% of TMZ an hour after intravenous administration, the overall decapsulation was delayed. In the next 24 h, it only averaged about 63% for PBCA-TMZ and PBCA-polysorbate-80 encapsulated TMZ [79] (Figure 3).

However, surfactants are rapidly degraded even before the BBB is reached with intravenous administration. Kudarha et al. worked on this problem by modifying albumin nanoparticles with TMZ by conjugating them with hyaluronic acid (HA-TMZ-NPs). The researchers achieved a significant accumulation of TMZ due to receptor interaction and caveolae-mediated endocytosis through BBB tissues [80,81,82,83]. According to the available data, there is a successful experience in using TMZ as part of supermagnetic nanoparticles, which showed a high degree of drug setting and encapsulation, as well as a long release period (90% of temozolomide was released for about 21 days). PLGA was used as the basis for supermagnetic nanoparticles with temozolomide. In addition, there are data that suggest a good potential of supermagnetic iron oxide particles [84,85].

Targeted transportation can be improved by attaching transferrin receptor (TF) ligands to TMZ-NPs located on the surface of the endothelial membranes of the brain capillaries. Receptor-mediated endocytosis makes it possible to transport temozolomide to a specific focus of interest to clinicians. CD44 receptors, which are actively expressed in glioblastoma tumor cells, belong to a similar “auxiliary tool”. Therefore, albumin nanoparticles connected to chondroitin sulfate (CD44 receptor ligand) have proved to be no less effective as TMZ transporters [86,87]. A comparative study of TMZ release from nanoparticles and nanoparticles conjugated with chondroitin sulfate showed a longer period of drug release from nanotransporters with CD44 receptor ligands, which amounted to 26%, 32%, and 42% at 2 h, 24 h, and 48 h, respectively [88]. Another example is a magnetic nanoparticle made of three polymers: Poly (ethylene Glycol)–Poly (Butylene Adipate)–Poly (ethylene Glycol) combined with folic acid. The three-component polymer confirmed the assumption of high efficiency in terms of high and specific accumulation of TMZ in glioblastoma C6 cells, mainly due to the folic acid ligand, whose receptors appear in increased quantities on the surface of tumor cells. The modification of nanoparticles with folic acid increases cellular uptake of drug transport systems by 2.5 times (*p* < 0.0001) [89].

In physiological conditions, TMZ undergoes rapid degradation due to its hydrolytic instability. This leads to the reduction in its half-life (1.8 h), and, consequently, to a decrease in the therapeutic effect and a subsequent higher frequency of drug administration. There are many strategies for TMZ stabilization, including its co-crystallization with organic acids, biocompatible polymer frameworks, and encapsulation in nanotransport systems [90,91,92]. Di Martino et al. created a transport system of amphiphilic nanoparticles based on chitosan and carboxy-enriched polylactic acid, which demonstrated stability both in an acidic (pH 5.5) and neutral (pH 7.4) environment [93].

### 4.4. Metal Nanoparticles

According to another research study, metal nanoparticles have the necessary properties to perform the role of transporters of therapeutic agents [94,95,96,97]. Liang et al. developed a complex of silver and temozolomide nanoparticles (AgNPs-TMZ). Even at very low concentrations (46 mmol/L), AgNPs showed high cytotoxic activity against glioma tumor cells [54].

Interestingly, zinc oxide nanoparticles with temozolomide synthesized from an aqueous extract of Glycyrrhiza glabra seeds demonstrated an IC50 of only 30 microg/mL. This is important for the mortality rate in glioblastoma cell lines [98].

### 4.5. Carbon Quantum Dots

A new treatment for brain tumors currently under development uses temozolomide incorporated into the structure of carbon quantum dots (CQDs), which are nanoscale clusters of 10 to 1000 atoms (<10 nm) of inorganic semiconductor crystals [99,100]. The product is based on a polymer biodegradable implant with chitosan-poly ethylene oxide-carbon quantum dots/carboxymethyl cellulose-polyvinyl alcohol (CS-PEO-CQDs/CMC-PVA). However, scientists faced the problem of early uncontrolled release of a large dose of the drug and the subsequent neurotoxicity. The problem was solved by coaxial electrospinning, which helped to create a transport system with a shell of carboxymethyl cellulose and polyvinyl alcohol, and chitosan and polyethylene oxide in its core. This increased the strength of the nanofibers and subsequently provided a more controlled and prolonged TMZ release [101,102,103].

### 4.6. Gene Therapy

Gene therapy is beginning to take an active leading position, offering to include nucleic acid genes in drug delivery systems [104]. Glioblastoma cells are able to develop drug resistance to temozolomide due to the O6-methylguanine-DNA-methyltransferase (MGMT) gene [105]. Zhu et al. constructed hybrid nanoparticles based on a lipid polymer with encapsulated clusters of short palindromic repeats associated with protein 9 (CRISPR/cas9)—LPHNs-cRGD. The surface of the transporters was paired with cRGD to bind to the avß3 integrin receptors in the MGMT gene of tumor cells. Exposure to the MGMT gene leads to a decrease in the precipitation of the MGMT protein, and subsequently glioblastoma cells restore sensitivity to temozolomide. For the successful delivery of the developed nanosystem through the BBB, the noninvasive focused ultrasound method was used (microvesicles acted as molecules that locally violate the permeability of the BBB) [104,105,106,107]. Finally, the results were obtained, demonstrating the effectiveness of transfection of 30–40%, especially in the case of a drug-resistant brain tumor. The LPHNs-cRGD system increased the survival rate of experimental mice with glioblastoma, which gives hope for the effective use of these transporters in clinical practice [108]. Tetrahedral nucleic acid (NP-TFNA) nanoparticles also demonstrated good results due to their ability to capture MGMT, and the modification of nanotransporters with GS24 aptamer made it possible to overcome the tight junctions of the BBB of mice [109]. Gene therapy is of high scientific interest and gives great hope in the treatment of brain tumors, but due to the complexity and high cost of its implementation in routine clinical practice, it is not yet quoted.

### 4.7. Nanocomposites and Nanotubes

In an effort to reduce the side effects of the antitumor drug, researchers have improved nanotransport systems in every possible way, creating multifunctional nanocomposites containing TMZ in a hydrazide form and conjugated with a polymer (β-L-malic acid), with a transferrin receptor adsorbed on the surface by a monoclonal antibody. pH-sensitive trileucine was added to overcome the endosomal destruction block in TMZ-resistant tumor cell lines. This modification caused a decrease in the viability of human glioma and human breast cancer cell lines (U87MG cells, MDA MB-468), and increased the half-life of TMZ by 3–4 times [110].

Hybrids of drugs such as TMZ in combination with gemcitabine (GEM) and decitabine (DAC) encased in gold nanoparticles have demonstrated cytotoxicity synergism against malignant glial cells U87. They stop the cell cycle in the G2/M phase by subjecting DNA to alkylation. According to research data, TMZ had a greater effect in combination with GEM [111,112,113].

It is worth mentioning carbon nanotubes consisting of hundreds of carbon concentric shells. These nanomaterials have good biocompatibility and a high surface area for loading the drug substance. At the moment, only physicochemical and biopharmaceutical properties of carbon nanotubes with TMZ included in their structure have been studied [114,115,116].

### 4.8. Researches of TMZ Encapsulated in Nanotransporters

TMZ encapsulated in nanoparticles has indeed demonstrated its effectiveness based on the results of several studies. For example, Abrudan et al. analyzed the properties of TMS enclosed in polymer nanotransporters with chitosan on highly differentiated cancer stem cells isolated from glioma. Evaluation of cytotoxicity and apoptosis in vitro showed the ability of the alkylating drug TMZ in combination with the nanosystem to change the resistance of tumor stem cells, which is just significant in the treatment of highly aggressive glioblastoma. Three types of nanotransporters were synthesized for the study (chitosan-TMZ, TMZ-chitosan-PEG (polyethylene glycol) and TMZ-chitosan-PPG (polypropylene glycol), which treated three types of cell lines (Glioma-derived stem, human fetal lung fibroblasts, HFL and human umbilical vein endothelial cell, HUVEC). The control method was a comparative analysis of survival in standard therapy with free TMZ and TMZ enclosed in nanocarriers. As a result, the authors obtained a large decrease in the survival rate of tumor cells when using TMZ encapsulated in chitosan and polymer nanoparticles than with traditional TMZ chemotherapy [117]. In their work, Jain et al. concluded that TMZ was released more slowly and that C6 glioma cells were highly absorbed using PLGA nanoparticles, which was evaluated using confocal microscopy and flow cytometry [118]. Another interesting study is the experience of using temozolomide-loaded albumin nanoparticles on rats. Animals were injected intravenously with TMZ nanoparticles obtained by high-pressure homogenization using bovine serum albumin. Parameters such as the time to reach the maximum concentration and absorption (the area under the “plasma concentration–time” curve from the moment of intake to the last detectable concentration at the time point t, AUC0→t) were significantly higher in comparison with the TMZ solution [119]. Thus, nanostructures can be more effective as vectors for drug transport than simple solutions of anticancer drugs, in particular, TMZ.

### 4.9. The Approaches to Fabricating Nanoscale Delivery Systems

Chemical methods for obtaining nanoparticles provide the proper level of purity, allowing us to obtain mixtures of several components to vary the morphology, crystal structure, and chemical composition of the resulting particles within a wide range. One of the limitations of these methods is the high degree of aggregation of particles and agglomerates. This problem is solved by using two main directions, which are used together or separately, depending on the specific task. The first direction is the creation of a certain stabilizing system in the reaction space that prevents the agglomeration of particles: the introduction of stabilizers of various mechanisms of action into the solution, the synthesis in a matrix that fixes particle sizes, the synthesis in liquid two-phase systems, etc. This group includes the sol–gel method, which involves the formation of a framework of molecules of an additional substance and ensures the homogeneity of the product at the molecular level. Polymer materials can be used to produce metal oxides. For example, in several works, polyacrylamide was used for these purposes [120].

The introduction of an additional substance at the synthesis stage has several significant drawbacks. Firstly, at the stage of heat treatment of powders, after removing additives, residual porosity may occur inside the particles. Secondly, additional components may remain in the product in the form of impurities, which is undesirable if it is necessary to obtain high-purity powders.

The second direction to prevent the aggregation of particles is the use of special installations that allow the removal of particles from the solution quickly, fixing them in granules or agglomerates in which the particles are weakly interconnected. In this case, it is possible not to introduce additional components into the initial solutions, except for precursors, or use them in smaller quantities. Many researchers have used the spray pyrolysis method (aerosol pyrolysis) for the synthesis of aluminum oxide particles.

To obtain nanoparticles, the spray drying process is also actively used, which is implemented in installations that allow the acquirement of spherical particles with nanometer-scale crystallite sizes at medium pressures less than atmospheric (~3000 Pa) and relatively low temperatures (~70–80 °C). Basically, research in this area is aimed at obtaining pharmaceutical ingredients, for example, encapsulated particles: cells, bacteria, food additives, etc.

Nano-spray drying plants are not designed for the production of nanopowders in large quantities. Powders obtained in this way can be used where reproducible morphology and purity of the product are necessary, for example, in the synthesis of drug carriers [120,121].

## 5. Conclusions

The use of nanotransporters for the targeted delivery of temozolomide to glioblastoma cells appears to be an impressive and promising opportunity to optimize the therapy of this malignant brain tumor. The size of nanoparticles and their modification by various ligands specific to receptors expressed on the endothelium of the brain capillaries makes it possible to successfully overcome the BBB. Encapsulation of an antitumor drug with subsequent controlled release and an extended half-life helps to reduce the toxic effect of TMZ on healthy tissues, protects temozolomide from premature physiological degradation due to MGMT, and delivers it directly to tumor cells. Advantages of using nanoparticles in comparison with conventional means of drug delivery: reduced toxicity (nanopreparations reduce the toxicity of the drugs contained in them), targeted delivery (nanoscale systems are used to deliver drugs to certain tissues and ensure the controlled release of active substances), and minimization of the use of excipients (solubilizers). All this will eventually lead to the introduction of nanomedicine into clinical practice and a breakthrough in the treatment of glioblastoma.

## Figures and Tables

**Figure 1 molecules-27-03507-f001:**
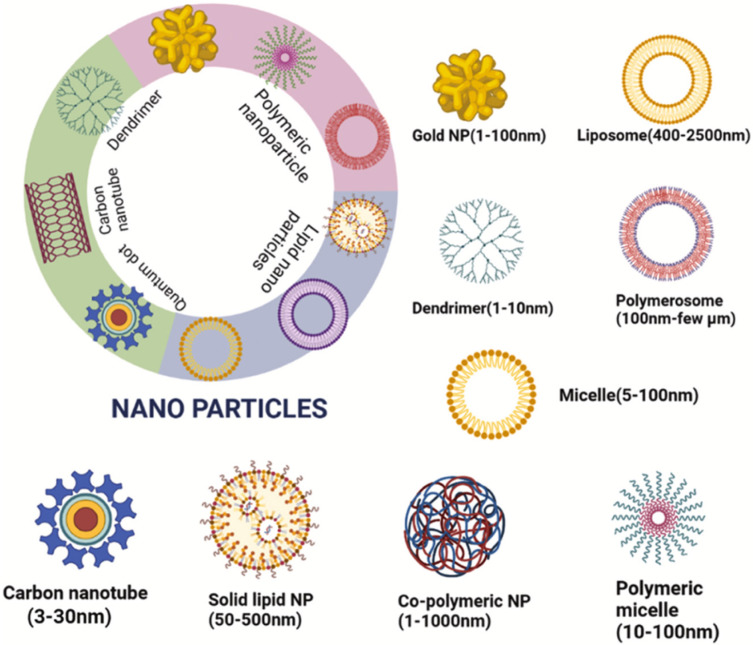
Various classes of nanocarriers [52].

**Figure 2 molecules-27-03507-f002:**
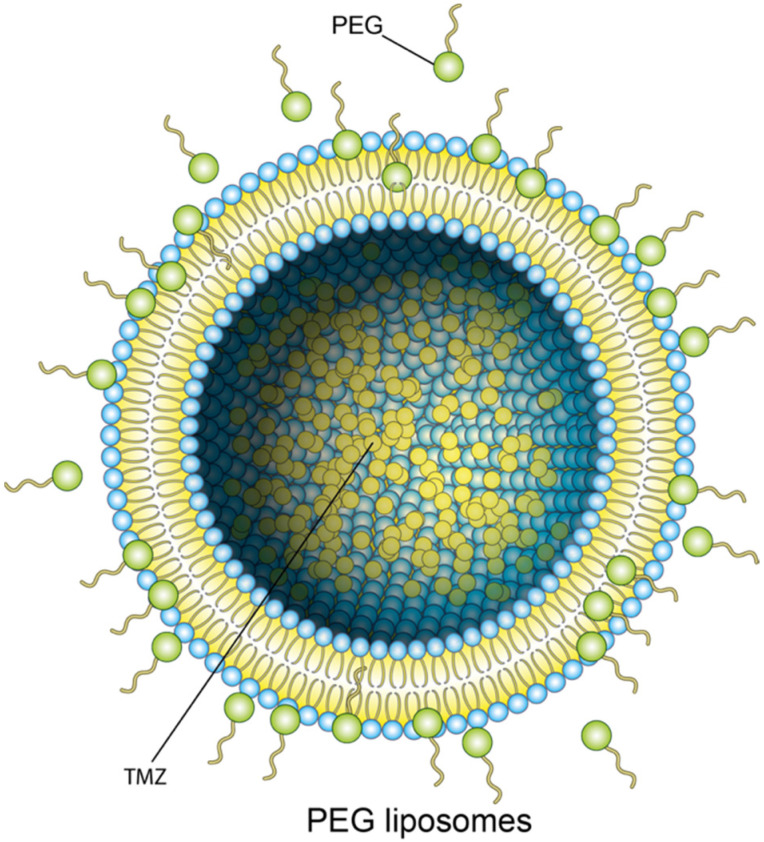
PEG liposomes.

**Figure 3 molecules-27-03507-f003:**
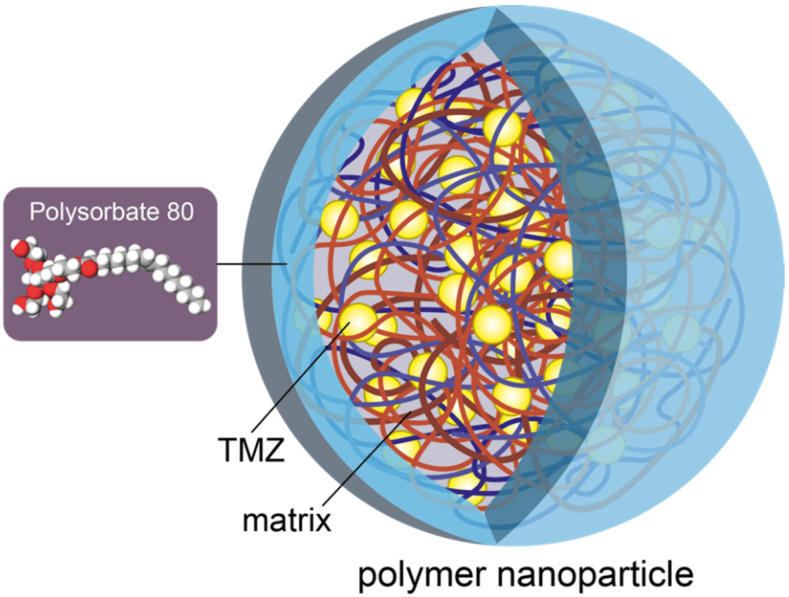
Polymer nanoparticle with polysorbate 80.

## Data Availability

The data presented in this study are available on request from the corresponding author.

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
