# Peer review of "Temozolomide Efficacy and Metabolism: The Implicit Relevance of Nanoscale Delivery Systems [Author-notes fn1-molecules-27-03507]"

_molecules, 2022, doi:10.3390/molecules27113507_

Round 1

Reviewer 1 Report

This review manuscript discusses the prospect of nanocarriers to potentiate the antitumor effects of temozolomide, a photoreactive alkylating agent. However, there is a serious problem for the publication in the journal, Molecular Structure. The most important deficit of this manuscript is just a composite of other researchers’ review articles. The manuscript does not afford the dynamism of the researchers who actually involved in the topic and is not likely to inspire further experimental studies that could lead to light new potential therapeutic roles of nanocarriers in brain tumors. As a scientific literature, more attractive and specific data should be added in terms of functional roles of the nanocarriers such as physical properties and results of animal experiments or clinical studies, even if they are someone else's data. To accept for publication, the manuscript should be revised thoroughly in regard to following issues.
Major comments
1. Not only stating data in writing, but also citing figures or tables of references under the reprint permission are needed.
2. Wording, sentence, and construction are inadequate and poor (for instance, obviously odd wording and sentences are marked in yellow in attached pdf). The manuscript should be refined thoroughly by authors first and edited by professional English editor at last.
3. Too many authors (line 4, 5, marked in orange in attached pdf). Description in “Author contributions (line 371)” is superficial. Omit authors who contribute little and specify each role in manuscript preparation.
4. Section 1, Introduction is concerned mostly with brain tumors in general and does not address the main topic, therapeutic prospect of nanocarriers. The general discussion should be deleted and changed to fit more the purpose of the article.
5. Section 4, Nanoscale delivery systems is too long. It should be divided into appropriate sections.
Specific comments
1. Line 12, the connection between glioma and glioblastoma is unclear.
2. Line 32, specify CBTRUS as Central Brain Tumor Registry of United States and cite URL.
3. Line 81,82, specify whether the number is mean or median.
4. Line 100, add – between O and 6.
5. With regard to the usability of nanocarriers, the authors only cite previous review articles and the rest is just their speculation. Discuss the usability by quoting more specific data.
6. In line 183 to 187, the reference number for Gabay et al is missing. Although the quoted contents seem interesting and critical, the description is too poor. Authors should mention detail of the work.
7. The form of references is rough and does not fit the instruction for authors at all (line 381-634).

Author Response

COVER LETTER TO THE REVIEWER

Dear reviewer, our team of authors has revised our work and the formats of articles proposed by the journal. We really agree with you that “PERSPECTIVE” is a more suitable format of work according to the criteria, and therefore we publish the manuscript in this category. Thanks for the feedback.

For a cover letter in which address the referees’ comments, please find attached

Kind regards,

Daria Petrenko

Reviewer 2 Report

In this manuscript, the authors reviewed the temozolomide efficacy and metabolism: the implicit relevance of nanoscale delivery systems. In my opinion, some issues should be further addressed and I hope the following comments could be helpful for improving their paper.

  1. In the introduction, the background about nanoscale delivery systems is little, the authors should enrich this part and emphasize the necessity of "Temozolomide efficacy and metabolism"
  2. Authors focused on Temozolomide efficacy and metabolism, but what are the distinguished properties and specific problems of  Temozolomide? The authors never discussed it.
  3. Good quality figures are very important for a good review paper, but I  find only two figures in this manuscript. Try to add at least 4-to 5 figures in this manuscript.
  4. The authors should summarize the current approaches to fabricating "nanoscale delivery systems" and compare their advantages and disadvantages in order to draw the reader's attention.
  5. This manuscript is well organized but lacks specific comparative analysis. What are the advantages of "nanoscale delivery systems"  compared with traditional technology?
  6. The author mainly discussed temozolomide efficacy and metabolism. It would be better if the nanoscale delivery systems could be elaborated more profoundly. Please cite some recent literature. https://pubs.rsc.org/en/content/articlelanding/2019/bm/c9bm00139e/unauth, https://pubs.rsc.org/en/content/articlehtml/2022/ma/d1ma00961c , https://pubs.rsc.org/en/content/articlelanding/2019/tb/c9tb01842e/unauth 
  7. Please revisit the entire manuscript for minor grammar issues.
  8. In conclusions and perspectives, the author should consider giving some methodological design about how to improve the performance of such "nanoscale delivery systems".

Author Response

Dear reviewer, we have changed the format of the article to "perspective",

For a cover letter in which address the referees’ comments, please find attached

3) therefore, the number of tables and figures have remained the same.  At the same time, we drew the structure of the nanoparticles ourselves, these drawings are original.  

1) The paper discusses in successive paragraphs the most practically applied and used forms of transport nanodelivery, in particular, those that have been investigated with the inclusion of temozolomide in them. We decided to leave the introduction more generalized.

2) The main advantage of temozolomide is its high effectiveness against malignant glioma of the brain, the main disadvantage is the safety profile. TMZ is a very toxic drug for surrounding tissues, therefore nanoparticles are considered as promising carriers capable of leveling this disadvantage, while increasing penetration through the blood-brain barrier and reducing the metabolic rate of TMZ.

4)-8) We have added data from literature sources for the last year. Discussing each type of nanotransporter, we describe what results, positive and negative sides were obtained during the research.

Thank you very much for the answer!

Kind regards,

Petrenko Daria

Round 2

Reviewer 1 Report

All comments have been addressed.

Author Response

Dear reviewer,

Thank you so much for your reply!